# Functional bottlenecks can emerge from non-epistatic underlying traits

Anna Ottavia Schulte[1], Samar Alqatari[2], Saverio Rossi[1], Francesco Zamponi[1]*

**1** Dipartimento di Fisica, Sapienza Università di Roma, Rome, Italy, **2** Department of Physics and The James Franck and Enrico Fermi Institutes, The University of Chicago, Chicago, Illinois, United States of America

* francesco.zamponi@uniroma1.it

## Abstract

Protein fitness landscapes frequently exhibit epistasis, where the effect of a mutation depends on the genetic context in which it occurs, *i.e.*, the rest of the protein sequence. Epistasis increases landscape complexity, often resulting in multiple fitness peaks. In its simplest form, known as global epistasis, fitness is modeled as a non-linear function of an underlying additive trait. In contrast, more complex epistasis arises from a network of (pairwise or many-body) interactions between residues, which cannot be removed by a single non-linear transformation. Recent studies have explored how global and network epistasis contribute to the emergence of functional bottlenecks - fitness landscape topologies where two broad high-fitness basins, representing distinct phenotypes, are separated by a bottleneck that can only be crossed via one or a few mutational paths. Here, we introduce and analyze a stylized model of global epistasis with an additive underlying trait. We demonstrate that functional bottlenecks arise with high probability if the model is properly calibrated. Furthermore, our results underscore that a proper balance between neutral and non-neutral mutations is needed for the emergence of functional bottlenecks.

## Author summary

A central challenge in the study of protein evolution is understanding how interactions between mutations influence evolutionary dynamics. These interactions, collectively known as epistasis, play a key role in shaping protein 'fitness landscapes', representing maps between amino acid sequences and their functional performance. However, the impact of epistasis on landscape ruggedness and accessibility remains a subject of debate. Recent experiments have revealed the existence of functional bottlenecks: regions of low fitness that restrict evolutionary transitions between proteins with different functionalities. While these bottlenecks represent a significant constraint on evolutionary outcomes and are often attributed to complex networks of interacting mutations, we demonstrate that they can arise in a much simpler setting. Using a

**Data availability statement:** All simulations and analyses were performed using scripts available at Zenodo (https://doi.org/10.5281/zenodo.18131102). Data were downloaded from previously published sources [27,37]. Specifically, raw sequencing data were retrieved from the Sequence Read Archive (SRA) under BioProject accession PRJNA560590.

**Funding:** This research has been supported by first FIS (Italian Science Fund) 2021 funding scheme (FIS783 - SMaC - Statistical Mechanics and Complexity to SR) from MUR, Italian Ministry of University and Research and from the PRIN funding scheme (2022LMHTET - Complexity, disorder and fluctuations: spin glass physics and beyond to SR) from MUR, Italian Ministry of University and Research. The funders had no role in study design, data collection and analysis, decision to publish, or preparation of the manuscript.

**Competing interests:** The authors have declared that no competing interests exist.

stylized model where fitness depends nonlinearly on an underlying additive trait, we show that functional bottlenecks do not require complex network epistasis. Instead, they can emerge from the inherent variability of the mutational effect distribution. Specifically, these bottlenecks arise when the mutational landscape is dominated by small, nearly neutral effects, but remains punctuated by enough strongly non-neutral mutations to create sharp fitness transitions. Our findings offer a new perspective on the constraints shaping protein evolution and highlight the critical role of mutational effect heterogeneity in determining evolutionary accessibility.

## 1 Introduction

Understanding how genetic variation translates into differences in reproductive success is a central challenge in evolutionary biology. An organism's fitness—defined as the number of offspring it produces—depends on how its phenotypic traits interact with the environment. These traits, in turn, are shaped by the organism's genotype, suggesting a hierarchical mapping from genotype to phenotype to fitness. By assigning a fitness value to each genotype, one obtains a *fitness landscape* [1–5], a conceptual tool that captures the topology of the space of evolutionary possibilities.

Epistasis is a distinctive property of fitness landscapes. Broadly speaking, it is defined as the context-dependence of mutational effects [6–38]. More precisely, it is defined as follows. Consider a genotype $\mathbf{a} = (a_1, a_2, \cdots, a_L)$ with an associated fitness $F(\mathbf{a})$. The genotype $\mathbf{a}$ could be a protein sequence of length $L$, with $a_i$ representing the amino acid at site $i$, or a nucleotide sequence, or a sequence of zeros and ones representing the presence/absence of a given mutation, gene, etc. Epistasis occurs when the fitness change $\Delta F_i(a \to b)$ due to substituting $a_i = a$ with $a_i = b$ at site $i$ depends on the amino acids that are present at other sites in the sequence. As such, epistasis can be of two conceptually distinct origins.

What is sometimes called 'global' epistasis is based on introducing the simplest possible non-linearity in the genotype-phenotype-fitness mapping [2,3,5,19,23,24,27,31,32,36,37,39]. More precisely, one assumes the existence of an underlying additive phenotype (or 'trait') $E(\mathbf{a}) = \sum_{i=1}^{L} h_i(a_i)$ associated with each genotype $\mathbf{a}$. Under this assumption, the variation of phenotype $E(\mathbf{a})$ associated with substituting $a$ with $b$ at site $i$, $\Delta E_i(a \to b) = h_i(b) - h_i(a)$, is independent of the rest of the genotype. Next, one assumes that the fitness $F(\mathbf{a}) = \phi(E(\mathbf{a}))$ is a non-linear function of $E(\mathbf{a})$, for instance a sigmoid (Fig 1A). Under these assumptions, while the variation of the underlying trait is independent of the rest of the sequence, the variation of fitness depends on it due to the non-linearity of $\phi$. Yet, such non-linearity is relatively easy to handle, because one can deduce the function $\phi(E)$ by 'deconvolution' of experimental fitness measurements [23,24,27,32,37,40–42], and thus describe the fitness function with a limited number of parameters. In concrete examples, few parameters enter in the definition of $\phi(E)$, and of the order of $L$ parameters enter in the definition of $E$. Furthermore, such a globally epistatic fitness landscape only features a single maximum, because $F = \phi(E)$ is an increasing function of $E$, and $E$ itself has a single

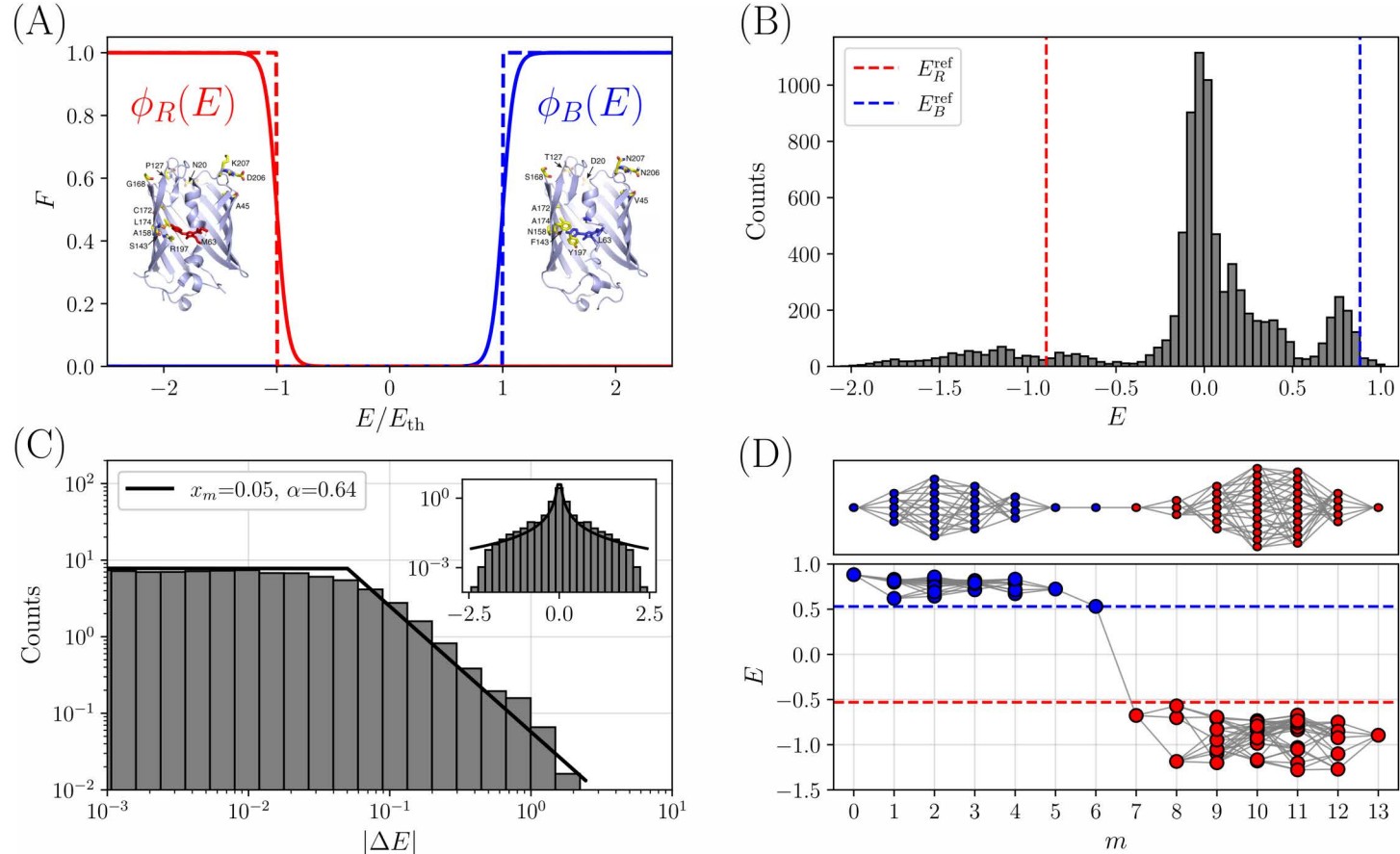

**Fig 1. Experimental data from Ref [27]. (A)** Schematic shape of the fitness functions for the red and blue phenotypes as a function of the underlying trait $E$. Sigmoid functions have been used for illustrations. The two reference structures with the corresponding mutations are also shown, from Ref [27]. **(B)** Histogram of the value of $E$ obtained from Eqs (2), (3) for each of the $2^{13}$ variants. The red and blue lines correspond respectively to the reference values $E_R^{\text{ref}}$ and $E_B^{\text{ref}}$. **(C)** Histogram of the absolute value $|\Delta E|$ of single mutational effects (SMEs) from Eq (5) for all $13 \times 2^{13}$ single mutations that can be obtained from the dataset, shown in log-log (main panel) and lin-log (inset) scales. The black curve is the fit obtained with the Pareto distribution in Eq (6). **(D)** Topology of the space of paths obtained keeping only genotypes with $|E(\mathbf{a})| > E_C$, and finding the largest possible value of $E_C = 0.53$ (dashed line) such that the red and blue reference sequences remain connected. The lower panel reports the values of $E(\mathbf{a})$ for each functional genotype (blue dots for $E > E_C$ and red dots for $E < -E_C$) as a function of the number of mutations from the blue reference, with the gray lines connecting pairs of genotypes that differ by a single mutation. The upper panel shows the resulting graph of connections.

maximum, obtained by choosing at each site the value $a_i$ that maximizes the additive contribution $h_i(a_i)$ [5]. Multi-peaked landscapes can be obtained by choosing a non-monotonic function $\phi(E)$, as in Fisher's Geometric Model [2,3,5,19,43].

A more complex behavior is obtained in presence of what is sometimes called 'network' epistasis [8,11,12,14,15,20,22,28–30,33–35,40–42,44], which implies that no function $E = \phi^{-1}(F)$ can transform fitness into a purely additive function of the genotype [27,32,37]. While a function $\phi$ can be chosen to make $E$ as close as possible to an additive form, higher-order interactions will still play a role in the presence of network epistasis. More formally, $E$ can always be decomposed as a sum of additive, two-site, three-site, and higher-order interactions [10,14,17,18,21,27,28,32,37], i.e.,

$$E(\mathbf{a}) = \sum_i h_i(a_i) + \sum_{i,j} J_{ij}(a_i, a_j) + \sum_{i,j,k} K_{ijk}(a_i, a_j, a_k) + \cdots$$

$$(1)$$

however, due to network epistasis, no choice of $\phi$ can eliminate the higher-order terms $J_{ij}$, $K_{ijk}$, and beyond. Let us focus for simplicity on the case in which only the two-site interactions are present. The number of parameters required to fully specify the function $E$ now scales as the number of pairs, i.e., proportionally to $L^2$, which makes inferring the fitness function from a few measurements significantly more difficult. Even more importantly, the function in Eq (1) can have many local maxima, corresponding to many fitness peaks, leading to a rough, complex fitness landscape. From Eq (1), any nonlinear function of $E(\mathbf{a})$ – that is, any form of global epistasis – can, in principle, be represented as a (possibly infinite) sum of high-order terms on the right-hand side. However, this representation is less practical, as it requires estimating a large number of parameters. A more effective strategy is to first capture as much of the nonlinearity as possible using a single global function. This substantially reduces the residual nonlinearity, making it easier to model the remainder with only low-order local terms.

The relative importance of global and network epistasis, and how to properly fit $E(\mathbf{a})$, has recently been the subject of intense debate [40–42]. A particular focus of the debate has been the role played by epistasis in shaping the topology of the fitness landscape in the proximity of functional switches. For instance, Poelwijk et al. [27] studied experimentally the fitness landscape separating two phenotypically distinct variants of the *Entacmaea quadricolor* fluorescent protein, one fluorescent in the red and the other in the blue, separated by 13 mutations. Assaying all $2^{13}$ intermediate variants, they observed, in addition to global epistasis, significant network contributions (both pairwise and higher order) to fitness. They also identified a functional bottleneck separating the 'blue' and 'red' space of fluorescent variants, i.e., a narrow region of accessible evolutionary paths between the two phenotypes, which they attributed to epistatic constraints. These functional bottlenecks illustrate how fitness landscapes can remain navigable despite their ruggedness. While epistasis can constrain viable evolutionary paths, it can also facilitate adaptation by carving out narrow high-fitness pathways that connect otherwise isolated functional states – typically via critical "switch" mutations [45–47].

In a subsequent study, Alqatari and Nagel [38] used similar methods to investigate functional switches in an elastic network, considered as a model for allosteric behavior in proteins. Analyzing ensembles of fitness landscapes, they observed different topologies at the threshold for viable evolution, and found that functional bottlenecks – often involving a single critical mutation – emerged generically, similar to the observations in the experimental study.

Many other recent studies investigated small combinatorial fitness landscapes. However, in most cases, these landscapes connect a reference wild type to an evolved variant with increased or decreased fitness with respect to a single functionality [37,48,49]. Experimental studies that investigate the combinatorial landscape separating two reference genotypes with distinct functionality are less widespread [32,50–52]. Moreover, in many of these studies functional transitions proceed through promiscuous intermediate variants that are as fit, or even fitter, than the two reference genotypes. Allowing for functional promiscuity represents a less constrained scenario than one where a functional switch must occur between mutually exclusive phenotypes. In the latter case, which is the one we consider here, the requirement for specialization naturally makes the landscape more prone to the formation of bottlenecks. The question of how widespread bottlenecks are – particularly in presence of promiscuity – is not addressed here.

In this work, we aim to identify the minimal conditions under which bottleneck topologies can emerge. Specifically, we investigate whether network epistasis is a *necessary* condition for the existence of functional bottlenecks in fitness landscapes. To address this, we introduce a simple, stylized model of global epistasis based on a random underlying additive trait and a nonlinear fitness function. Our model is not intended to fit specific empirical data; rather, it follows the spirit of the Fisher's Geometric Model and related frameworks [2,3,5,19,43] by generating an ensemble of random fitness landscapes with stylized features. Upon calibration, we demonstrate that our model encompasses a wide range of parameters for which the resulting topologies exhibit a functional bottleneck with high probability. This result, by itself, shows that global epistasis alone is sufficient to generate functional bottlenecks.

Furthermore, we studied the role of the distribution of single mutational effects (SMEs). More specifically, we find that when the model is properly calibrated, the distribution of SMEs selected by evolution displays a proper balance between

nearly neutral SMEs and enough strongly non-neutral SMEs, which is needed to create sharp fitness transitions. Our stylized model thus sheds light on some fundamental constraints that shape fitness landscapes and possibly evolution.

## 2 Results

The central result of our work is the introduction of a stylized fitness landscape model, designed to investigate the minimal conditions under which bottleneck topologies emerge. However, to provide the motivation and inspiration for this construction, we begin with a pedagogical discussion of experimental data from Poelwijk et al. [27]. This analysis of empirical data serves to ground the stylized features of the model that follows.

### 2.1 Experimental data on fitness bottlenecks

Poelwijk et al. [27] analyzed a family of variants of the naturally occurring wild type *Entacmea Quadricolor* red fluorescent protein *eqFP611* [53]. More specifically, from a natural variant of *eqFP611*, called *eqFP578* (with 76% amino acid sequence homology), the *TagRFP* mutant was engineered via random mutagenesis. *TagRFP* is 21 mutations away from *eqFP578* [54, see S1 Text] and has enhanced fluorescence and stability. From *TagRFP*, two engineered variants have been derived to have distinct fluorescence (See https://www.fpbase.org/protein/eqfp578 for the complete tree of variants.):

• *mTagBFP2*, which is fluorescent in the blue and has 13 mutations and 6 insertions relative to *TagRFP* [55];

• *mKate2*, which is fluorescent in the deep red and has 8 mutations and one insertion relative to *TagRFP* [56].

The sequences of *mTagBFP2* and of *mKate2* differ by 13 mutations (12 of which are shown in Fig 1A reproduced from Ref [27]), thus resulting in a total of $2^{13}$ = 8192 different combinations of intermediate amino acid substitutions. The experiment of Poelwijk et al. [27] measured the fluorescence of all these intermediates, both in the red and in the blue. Note that there are also 5 insertions in *mTagBFP2* with respect to *mKate2* that were neglected in the experiment (details were not found in Ref [27]).

We focus on these published data as they characterize a combinatorial landscape bridging two reference genotypes with distinct functionalities, within which a functional bottleneck was explicitly identified. A more recent study, employing a similar experimental design but investigating a different protein, reported the absence of such a bottleneck [50]. This discrepancy highlights the topological variability of fitness landscapes and motivates our search for the minimal conditions that govern the emergence of these constraints.

We downloaded the original data from Ref [27] and represented the genotypes with binary vectors of 13 variables, with $\mathbf{a}_B$ = (0, 0, · · · , 0) being the blue *mKate2* and $\mathbf{a}_R$ = (1, 1, · · · , 1) being the red *mTagBFP2*. In the experiment, *E.Coli* cells expressing the mutant genotypes were sorted by a microfluidic device able to push them into separate channels according to their fluorescence. From the raw experimental measurement, we assign to each genotype a red fitness, $F_R(\mathbf{a})$, and a blue fitness, $F_B(\mathbf{a})$, based on the enrichment in the two channels, each normalized to its corresponding reference value such that $F_R(\mathbf{a}_R) = F_B(\mathbf{a}_B) = 1$ (see the S1 Text for details). This normalization provides a consistent reference scale, with each reference variant serving as the baseline functional state for its phenotype, enabling a more direct quantitative comparison between the red and blue fitness values.

Note that a specific measurable trait – fluorescence intensity – is here employed as a proxy for fitness, despite its lack of direct coupling to reproductive success. Instead, fluorescence quantifies the protein's functional efficiency. Under the assumption that enhanced biophysical performance may confer a selective advantage within the relevant environment, fluorescence serves as a proxy for fitness. This approach is standard in many empirical studies of fitness landscapes, such as Refs [27,50].

For each measured phenotype (we will use a suffix 'R' for red and 'B' for blue fluorescence), we can define an underlying trait and a non-linear mapping to fitness $F = \phi(E)$. Inverting the non-linear function, we can thus derive the underlying trait from the measured fitness,

$$E_R(\mathbf{a}) = \phi_R^{-1}\left[F_R(\mathbf{a})\right] \ , \quad E_B(\mathbf{a}) = \phi_B^{-1}\left[F_B(\mathbf{a})\right] \ . \tag{2}$$

The optimal methodology for inferring the function $\phi(E)$ remains a subject of active debate [24,32,37,40–42]; for instance, it has been suggested that the nonlinearity be inferred directly through rank-based statistics [42]. While acknowledging that results can be sensitive to the specific choice of $\phi$, for our illustrative purposes we keep the same power-law form $\phi^{-1}(x) = x^{0.44}$ utilized in Ref [27] for both phenotypes. Alternative specifications, such as the sigmoid function shown in Fig 1A, may yield quantitatively different results, but the qualitative topology of the landscape remains robust.

We note that when there are two phenotypes only, for simplicity, we can follow Alqatari and Nagel [38] and encode them both in a single trait

$$E(\mathbf{a}) = E_B(\mathbf{a}) - E_R(\mathbf{a}) \ , \tag{3}$$

which is positive for the 'blue' phenotype and negative for the 'red' phenotype. This is only possible when the two phenotypes are exclusive, as in Ref [27], which may not be true in other cases [32,50–52]. We also define the reference trait values as

$$E_B^{\mathrm{ref}} = E(\mathbf{a}_B) \ , \qquad E_R^{\mathrm{ref}} = E(\mathbf{a}_R) \ . \tag{4}$$

The structure of the two proteins and the 13 substitutions are illustrated in Fig 1A, together with a schematic illustration of the non-linear mapping between $E = E_B - E_R$ and the two fitnesses.

Fig 1B shows the histogram of $E(\mathbf{a})$ across all $2^{13}$ genotypes, with red and blue lines corresponding to the two reference variants. Note that the reference values are $E_B^{\mathrm{ref}} \sim 0.88$ and $E_R^{\mathrm{ref}} \sim -0.89$. Their absolute value slightly differs from one due to the contribution from the other phenotype. As observed in Ref [27], most genotypes are non-functional with $E \sim 0$. There is a rather sharp peak of blue genotypes around $E \sim 1$, while red genotypes exhibit a much broader distribution, ranging from $E \sim -0.5$ down to large negative values of $E$. This indicates that some intermediates are significantly more fluorescent in the red than the red reference variant itself.

Having defined $E(\mathbf{a})$, we can analyze the distribution of SMEs. For each of the $2^{13}$ backgrounds $\mathbf{a}$, we consider all 13 single mutations and we compute the SME,

$$\Delta E_i(\mathbf{a}) = E(a_i \to 1 - a_i \,|\, \mathbf{a}) - E(\mathbf{a}) \ . \tag{5}$$

The histogram of the absolute value of the $13 \times 2^{13}$ SMEs is reported in Fig 1C. (Note that the $13 \times 2^{13}$ SMEs include each mutation and its reverse, hence the histogram would be symmetric by construction, which is why we focus here on $|\Delta E_i|$.) Although we have limited data, we clearly observe that the distribution is fat-tailed. A good fit is achieved by a Pareto distribution,

$$p(x) = \frac{\alpha}{2x_m(\alpha+1)} \times \begin{cases} 1 & \text{if } |x| < x_m \ , \\ \left(\frac{x_m}{|x|}\right)^{\alpha+1} & \text{if } |x| > x_m \ , \end{cases} \tag{6}$$

here with $x = \Delta E$. The fit is just an indication, in particular due to the limited amount of data and the limited range of the experimental fitness, which both cutoff the distribution at large $\Delta E$. While other distributions could also fit the data, the distribution tails are quite broad, indicating substantial heterogeneity among SMEs (in the S1 text we show similar results for another, independent dataset of SMEs [37]).

Finally, in Fig 1D, we examine the topology of the resulting functional bottleneck. To perform this analysis, we need to first establish a formal definition of a 'functional' genotype. As noted in Ref [27] and illustrated in Fig 1B, the threshold for

functionality is inherently context-dependent; shifting this threshold can substantially alter the perceived landscape topology [38]. Indeed, viability is not an intrinsic property of a genotype alone but is contingent upon the environment – specifically the selection pressure exerted on the relevant trait – which varies across different experimental conditions. In this work, we consider that the fitness function is growing fast around a given threshold $E_{th}$ (either on the positive or negative side, see Fig 1A). Hence, genotypes with $|E(\mathbf{a})| < E_{th}$ are considered dysfunctional and evolutionarily disallowed. Because the reference variants are functional, they must of course be located above the functionality threshold, i.e., $|E^{ref}| > rsimE_{th}$. However, Fig 1D shows that no path of single mutations exists that connects the two reference variants while always maintaining $|E(\mathbf{a})| \geq E^{ref}$, hence we need to consider $E_{th} < E^{ref}$ in order to preserve the connection between the reference sequences. (Note that a different result was obtained in Ref [27] when the two phenotypes were not normalized to the reference values. We discuss this point in the S1 Text.) What is, then, the largest value of the functionality threshold $E_{th}$ such that at least one viable mutational path exists between the two reference variants?

In order to precisely answer this question, following Ref [38], we characterize the space of the paths connecting the two reference variants upon increasing the threshold $E_{th}$. We consider the subset of paths that (i) are made of single mutations, (ii) connect the red and blue reference sequences, and (iii) are such that $|E(\mathbf{a})| > E_{th}$ all along the path. We start from $E_{th} = 0$, and gradually increase $E_{th}$ to the maximum value such that at least one path remains. The resulting value, which we call $E_C$, quantifies how 'hard' it is for evolution to find a path that realizes the functionality switch. If $E_C \sim E^{ref}$, then one can find a path of single mutations, such that all intermediates are equally functional to the reference variants. If instead $E_C \ll E^{ref}$, in order to connect the two reference variants one has to accept a loss of fitness with respect to the reference fitness, which makes the transition less likely, but still possible.

For the experimental data of Ref [27], this analysis yields a value of approximately $E_C \sim 0.53$, or $E_C/E^{ref} \sim 0.6$. The resulting space of paths, depicted in Fig 1D, exhibits the characteristic bottleneck shape also reported in Ref [38], where all paths traverse a single 'jumper' genotype with $E = E_C$. In Fig 1D, the jumper genotype has a blue phenotype, after which a single mutation brings to the red phenotype. This functional switch is driven by a mutation located roughly at the midpoint. The number of functional intermediates determines the number of evolutionary paths that can reach the critical 'jumper' genotype. More paths imply greater evolutionary accessibility, making it easier for evolution to find the key mutation; while fewer paths indicate a more constrained transition requiring a specific mutational sequence. Note that the analysis in Ref [27] found a less pronounced bottleneck, with multiple mutational paths surviving – a scenario that can be reproduced here by adopting a less stringent definition of $E_C$ which allows for multiple paths, see Ref [38].

## 2.2 Stylized model of global epistasis

The goal of this paper is to determine whether a bottleneck structure like the one observed in Sec. 2.1 requires network epistasis, or if it can also be explained by a simpler model of global epistasis.

To formulate a simple stylized model, we begin by an important observation. The reference proteins analyzed in Ref [27] are not the product of natural evolution, but rather highly engineered sequences. In engineered proteins, derived either by random mutagenesis as in Sec. 2.1 or by directed evolution, one typically observes that the fitness improvement is achieved by just a few highly beneficial mutations, see, e.g., [37,57,58]. While this might seem, in principle, a highly specific feature of engineered proteins, a similar scenario has been observed when a natural protein needs to quickly adapt to a novel environment and acquire a new function. A notable example is the SARS-COV-2 Spike protein that displayed a few highly beneficial mutations just after the virus jumped to human hosts (see, e.g., [59]), but similar dynamics has been observed in other viral proteins [60], and in the immune system [32]. In each of these cases, just a few advantageous mutations allow the protein to acquire the new desired function, whether the selection pressure is applied artificially or emerges from a change in the natural environment. A very different dynamics might be at play in the neutral evolution

of optimized initial proteins [61–64], which might explain why the combinatorial landscape of Ref [50] does not display a bottleneck.

To capture a few minimal ingredients inspired by the results of Sec. 2.1 and the above discussion, we introduce a stylized model featuring only global epistasis. The model is defined as follows:

- The genotype $\mathbf{a} = (a_1, a_2, \cdots, a_L)$ is a binary sequence (i.e., $a_i \in \{0, 1\}$) of length $L$.

- An underlying additive trait $E(\mathbf{a}) = \sum_{i=1}^{L} h_i a_i$ has random SMEs $h_i$ that are identically and independently distributed according to a symmetric input distribution $P(h) = P(-h)$. After having drawn the values $h_i$ independently, we impose that $\sum_{i=1}^{L} h_i = 0$ by shifting their mean, i.e., $h_i \to h_i - \frac{1}{L} \sum_{i=1}^{L} h_i$.

- The two fitness functions that will be associated with the two 'colors' are non-linear functions of the underlying trait, $F_B = \phi_B(E) = \phi_0 / (1 + e^{\beta(E_{th} - E)})$ and $F_R = \phi_R(E) = \phi_0 / (1 + e^{\beta(E_{th} + E)})$, as illustrated in Fig 1A. If $E(\mathbf{a}) > E_{th}$ then the genotype $\mathbf{a}$ is functional 'blue', if $E(\mathbf{a}) < -E_{th}$ it is functional 'red', and if $|E(\mathbf{a})| \ll E_{th}$, the genotype is non-functional.

The fitness functions grow sharply around $\pm E_{th}$, in a way controlled by the parameter $\beta$. The choice we will make implicitly, following Ref [38], is to consider the limit $\beta \gg 1$, in which the fitness is close to a threshold (or Heaviside) function at $\pm E_{th}$ (see Fig 1A, dashed curves). Note that the values $h_i$ can be multiplied by an arbitrary constant that can be absorbed into $E_{th}$ and $\beta$, which allows us to fix the overall scale (e.g., the variance) of $P(h)$ without loss of generality.

Given the above definitions, both genotypes $\mathbf{a} = \mathbf{0} = (0, 0, \cdots, 0)$ and $\mathbf{a} = \mathbf{1} = (1, 1, \cdots, 1)$ have $E = 0$ and are thus non-functional. We stress that, contrarily to the analysis of experimental data reported above, these two genotypes will not be the reference sequences in the stylized model. Following Ref [38], we thus introduce a 'tuning procedure' to generate two reference variants, the 'red' $\mathbf{a}_R$ with $E_R^{ref} < -E_T$ and the 'blue' $\mathbf{a}_B$ with $E_B^{ref} > E_T$, for a chosen 'tuning' value $E_T > 0$.

The procedure starts from the genotype $\mathbf{a} = \mathbf{0} = (0, 0, \cdots, 0)$, which is thus considered as a 'common ancestor' or 'wild type' (for example, in the experiment described in Sect 2.1, it would correspond to the *eqFP578* protein). One then sequentially introduces mutations as follows.

- With probability $p$, we perform a 'greedy' step. We scan all the $h_i$ values that have not been introduced yet (those for which $a_i = 0$), and choose the one that has the maximum impact towards the desired goal, i.e., the largest positive $h_i$ for the blue reference variant or the smallest negative $h_i$ for the red reference variant. We introduce the corresponding mutation by switching $a_i = 1$.

- With probability $1 - p$, we perform a 'random' step. We choose at random one site $a_i$ such that $a_i = 0$, and mutate it to $a_i = 1$.

The tuning procedure is performed independently for the two colors starting from $\mathbf{a} = \mathbf{0}$, and continues until for the first time $E_R^{ref} < -E_T$ to construct the red reference variant, or $E_B^{ref} > E_T$ to construct the blue reference variant.

This procedure is not designed to be a fully realistic model of molecular evolution. Nonetheless, it is grounded in experimental observations. As noted at the beginning of this section, directed evolution experiments often yield final protein variants carrying a small number of beneficial mutations together with a few neutral ones. More generally, when a protein is challenged to acquire a new function, adaptation often involves selecting a handful of advantageous substitutions on top of incidental neutral changes. Our approach for constructing the reference variants is simply the most straightforward way to reproduce this pattern, without any claim of population-genetic realism. We also believe that this procedure mimics the tuning of elastic networks performed in Ref [38], in which the network is tuned based on a local proxy for the global fitness; we hereby assume that this local proxy can indeed provide the best choice (with probability $p$) or miss it (with probability $1 - p$).

The parameters that define the model are the sequence length $L$, the *a priori* distribution of SMEs $P(h)$, the probability $p$ and the tuning parameter $E_T$ that controls the reference sequence construction. We note that $P(h)$ may be interpreted as describing the a priori distribution of mutational effects – arising, for instance, from underlying biophysical constraints –

which represents the potential 'pool' of mutations accessible to evolution. We found that various choices of $P(h)$ lead to similar outcomes, provided the model is appropriately calibrated. For the sake of parsimony, and recognizing that these are stylized representations, we restrict our analysis to two distributions that capture distinct mutational behaviors.

1. **Gaussian**: A Gaussian distribution with unit variance.

2. **Pareto cutoff**: Motivated by the analysis of experimental data in Fig 1C, a Pareto distribution, i.e., Eq (6) with $x = h$, where we fix $\alpha = 0.7$ and $x_m = 0.1$. We also include an upper cutoff as in the data, i.e., we set $P(h) = 0$ for $|h| > 2$ and adjust the normalization constant accordingly.

For a given choice of parameters, an 'instance' of our random model is thus defined by the *a priori* pool of the $L$ SMEs, $h_i$, and by the two reference variants $\mathbf{a}_R$ and $\mathbf{a}_B$. For a given instance, we can define the number of mutations $M$ separating $\mathbf{a}_R$ and $\mathbf{a}_B$ (i.e., the number of $a_i$ that differ in the two sequences), and we can then investigate the $M!$ directed paths that connect them, obtained by introducing these mutations in all possible orders. Following Ref [38] and as discussed above, we can find the maximum value of $E_{\text{th}}$ such that at least one single-mutational path connecting the two reference variants, with all intermediate states having $|E| > E_{\text{th}}$, survives, and we call this value $E_C$. The closer $E_C$ is to $E^{\text{ref}}$, the easier it is to perform the functional switch.

Given $(L, p, E_T)$ and $P(h)$, we wish to characterize the statistical properties of the quantities $M$, $E^{\text{ref}}$, and $E_C$, which characterize the overall topology of the fitness landscape. Another interesting quantity is the distribution $P(\tilde{h})$ of SMEs $\tilde{h}_i$ selected by the tuning procedure to generate the two reference variants (i.e., those $h_i$ that correspond to $a_i = 1$ in each reference genotype), which may be interpreted as the distribution of mutations that are 'fixed' by evolution. We will also define $E^{\text{ref}}_{\text{max}} = \max(E^{\text{ref}}_B, |E^{\text{ref}}_R|)$. In the following, we will use brackets, $\langle A \rangle$, to denote the statistical average of an observable $A$ over an ensemble of random instances generated by the model. Unless otherwise specified, the average is taken over 20000 independent instances of the model for each value of the parameters.

Our model is similar in spirit to Fisher's geometric model (FGM) as discussed in Refs [2,3,5,19,43], from which it however differs in two important aspects. First, the choice of non-linearity, which reflects here the existence of two distinct fitness functions associated to each color, see also Ref [43] for a similar choice. Second, the fact that we choose two reference variants using the stochastic procedure described above, and we then restrict to the space of intermediates between the selected variants. As we show in the following, these two ingredients, together with a proper calibration of the model, give rise to the desired bottleneck structure.

Before discussing the details of the calibration procedure, a typical example of the space of mutational paths obtained from a calibrated model is shown in Fig 2. The qualitative similarity with Fig 1D, modulo the (irrelevant) change of scale of $E$, is striking.

### 2.3 Calibration of the model

The first step is to calibrate the parameters $(L, p, E_T)$ in such a way that our stylized model reproduces the basic phenomenology observed in Refs [27,38]. The requirements are the following.

- The phenotype of the two reference sequences should be comparable to $E_T$, i.e., $|E^{\text{ref}}|/E_T \simeq 1$, in order for $E_T$ to be a meaningful fitness scale.

- We want to reproduce a 'bottlenecked' structure of the space of evolutionary paths connecting the two reference variants, as shown in Figs 1D and 2 and in Ref [38]. This is characterized by a single 'jumper' genotype through which all paths must pass. The mutation responsible for the functionality switch occurs just after (or just before) this genotype. We note that the jumper genotype can be connected by a single mutation to only one, or more than one, genotypes carrying the other phenotype.

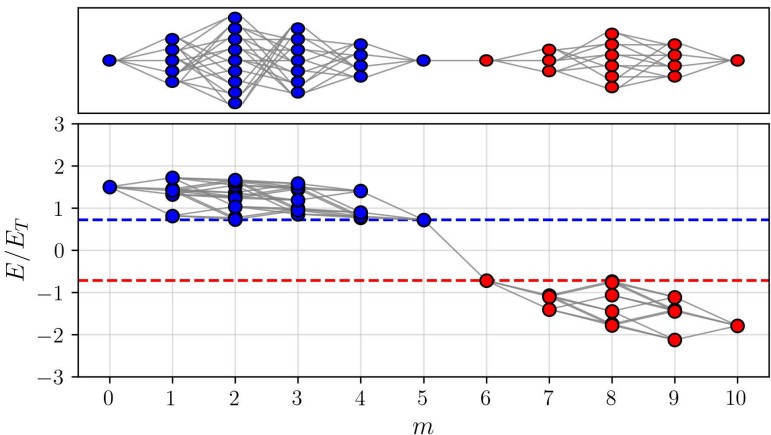

**Fig 2. Same representation as in. Fig 1D, using an instance of the calibrated model with Gaussian _P(h)_ instead of the experimental data.** The lower panel reports the value of $E/E_T$ for each functional variant as a function of the distance from the blue reference variant (here with $E_B^{ref}/E_T \approx 1.50$ and $E_R^{ref}/E_T \approx -1.79$). The threshold value for which at least a mutational path remains is $E_C/E_T \approx 0.72$ (dashed lines) and when normalized with respect to the largest of the two reference genotypes it reads $E_C/E_{max}^{ref} \approx 0.40$.

- The 'jumper' genotype should occur approximately at half distance along the evolutionary trajectory, as it is observed in experimental data (Fig 1D). Furthermore, we want the largest possible number of functional intermediate variants to survive before and after the jump, thus maximizing the number of functional evolutionary paths connecting the two reference variants.

- The fitness variation produced by the 'jumper' mutation(s) should be as large as possible, in such a way that $E_C$ is as close as possible to $E^{ref}$. If this is the case, a path can go in a single jump from having fitness close to the red reference, to having fitness close to the blue reference. Otherwise, one would necessarily have to tolerate a decrease in fitness to perform the functional switch. This makes the resulting mutational paths more viable for evolution.

- Finally, for practical reasons, the number of mutations $M$ connecting the two reference variants should be small enough, otherwise enumerating all the $M!$ paths becomes computationally very expensive. To avoid this, we impose $\langle M \rangle \sim 8$, a value close to that chosen in Ref [38].

An important remark is in order about the expected value of $E_C$. Consider for illustration the case in which the tuning procedure reaches the target with a single greedy step for both colors. We can thus expect $E_B^{ref} \sim \max(h_i)$ and $E_R^{ref} \sim \min(h_i)$. In absence of network epistasis, the largest jumper mutation that can happen along a path has $|\Delta E| \sim \max(|h_i|) \sim E_{max}^{ref}$, but this mutation must bring the system from $E_C$ to $-E_C$, hence $2E_C \sim |\Delta E| \sim E_{max}^{ref}$. Based on this simple argument, the largest possible value of $E_C$ we can expect is such that $E_C/E_{max}^{ref} \sim 1/2$, as we observe numerically (Fig 3C).

To fix the model parameters $(L, p, E_T)$ we proceed as follows. First of all, we fix $L = 500$ as the length of a typical protein. We checked that the results are very weakly dependent on $L$ (see the S1 Text). Next, for each $p$, we progressively increase $E_T$ until the average number of mutations $\langle M \rangle$ separating the two reference variants is $\langle M \rangle \sim 8$. The resulting $E_T(p)$ is plotted as a function of $p$ for both choices of $P(h)$ in Fig 3A, and we find that it is approximately linear in $p$. The linear fit allows us to fix $E_T(p)$ for all $p$.

The next step is to calibrate $p$. For this, we consider the ratio $|E^{ref}|/E_T$ for both colors, which we want to be close to one. The evolution of its statistical mean with $p$ is shown in Fig 3B. For small $p$, we mostly perform random steps in the training, and the requirement that $M$ is small forces $E_T$ to be small as well (Fig 3A). As a result, it is very likely to 'overshoot' the target and end up with $|E^{ref}|/E_T \gg 1$. Hence, in order to have $|E^{ref}|/E_T \sim 1$, we need a large enough $p$.

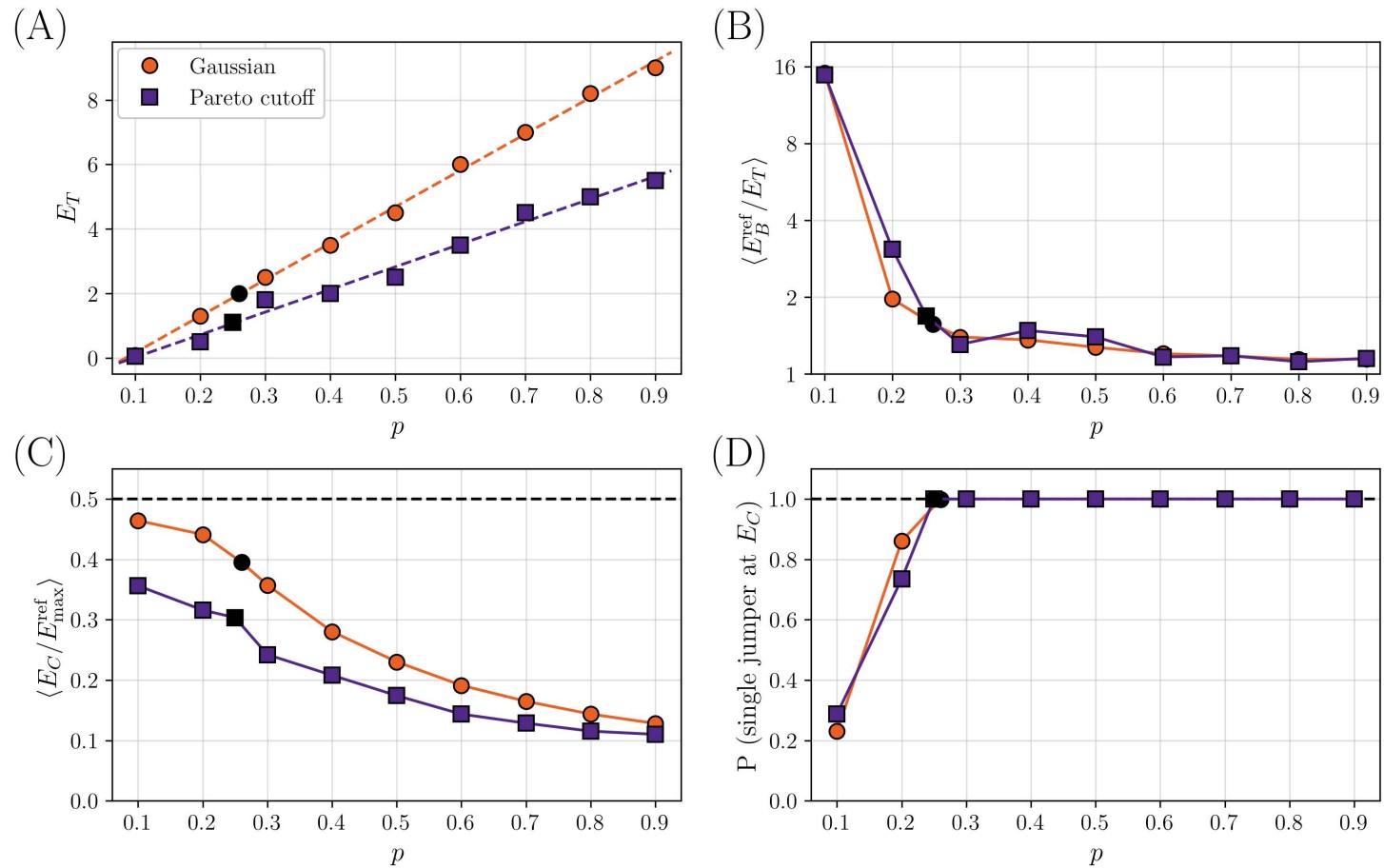

**Fig 3. Calibration of the two models with different $P(h)$, a Gaussian distribution or a Pareto distribution with cutoff.** The black symbols correspond to the choice of $p$ after calibration. **(A)** Value of $E_T$ for which $\langle M \rangle \approx 8$ as a function of $p$. Dashed lines are linear fits. **(B)** Average value $\langle E_B^{\mathrm{ref}}/E_T \rangle$ as a function of $p$. **(C)** Average of $E_C/E_{\max}^{\mathrm{ref}}$ as a function of $p$. **(D)** Probability of having a single jumper at $E_C$, or equivalently that $E_C < E_{\min}^{\mathrm{ref}}$, as a function of $p$.

Next, we consider the connectivity threshold $E_C$. We recall that $E_C$ is obtained, for each instance, in such a way that at least one single-mutational path connecting the two reference variants exists, with all intermediate states having $|E| > E_C$. Because both reference sequences must always be included in the space of allowed genotypes, increasing the value of $E_C$ beyond $E_{\min}^{\mathrm{ref}} = \min(E_B^{\mathrm{ref}}, |E_R^{\mathrm{ref}}|)$ is not meaningful [38]. This gives an upper bound for the largest value of $E_C$ for each instance. We show in Fig 3C its average value as a function of $p$, where the average is restricted to instances such that $E_C < 0.9E_{\min}^{\mathrm{ref}}$. When $p$ is too large, the tuning procedure requires too many greedy steps and as a consequence $\langle E_C/E_{\max}^{\mathrm{ref}} \rangle$ tends to decrease, contrarily to what we want in order to have bottleneck topologies with large $E_C$. This suggests to lower the value of $p$ as much as possible.

However, in Fig 3D we report the probability of having a single jumper at $E_C$, which we estimate for numerical convenience by the probability that $E_C < 0.9E_{\min}^{\mathrm{ref}}$, as a function of $p$. When $p$ is too small, this probability drops quickly below one. This is because $|E^{\mathrm{ref}}|/E_T$ starts increasing, the reference variants are obtained by many random steps, and the bottleneck structure is lost.

Taken together, these results suggest to choose the lowest value of $p$ such that the probability of a single jumper is still very close to one, which also gives both $E^{\mathrm{ref}}/E_T \sim 1$ and $E_C/E_{\max}^{\mathrm{ref}} \sim 0.5$. The value of $E_T(p)$ is fixed by the requirement that

$\langle M \rangle \sim 8$. For both choices of $P(h)$, this optimal value is around $p = 0.25$, although slightly larger values could be considered as well.

The crucial insight from the calibration is that one needs to tune $p$ in order to achieve a good balance between greedy and random steps. Correspondingly, as we will discuss later, the distribution $P(\tilde{h})$ of the SMEs that are selected by the tuning procedure that generates the reference sequences (that may be interpreted as those 'fixed by evolution') is very heterogeneous, featuring a coexistence of large SMEs (generated by greedy steps) and small SMEs (generated by random steps).

## 2.4 Results for the calibrated model

In the following, we choose $L = 500$ and the parameters corresponding to the black dots in Fig 3 as the final calibrated parameters for which we report more detailed results: Gaussian $P(h)$ with $p = 0.26$ and $E_T = 2.0$ for the tuning procedure, and Pareto cutoff $P(h)$ with $p = 0.25$ and $E_T = 1.1$. Keeping these two models fixed, we generated many ($\sim 20000$) random instances of the model and studied the distribution of the relevant quantities that characterize the topology of the fitness landscape. An example of a 'good' topology, closely resembling the one found experimentally, has already been given in Fig 2. We want to quantify the probability of generating such a topology in the stylized model.

We first focus on the properties of the two reference variants, shown in Fig 4 for both models. More specifically, Fig 4A shows the histograms of $M$, the total number of mutations separating the two reference sequences. We note that, by construction, $M \geq 2$, and we fixed $\langle M \rangle = 8$. We see that the distribution is quite broad, leading to a significant fraction of instances with $M$ as large as 20, and resembles that shown for elastic networks in Ref [38]. However, when we investigate the mutational paths, we restrict the analysis to instances with $M \leq 14$ for computational tractability, as in Ref [38]. Fig 4B shows the histograms of $E_B^{\text{ref}}/E_T$ for the blue reference variant (similar results are obtained for the red reference variant due to the inherent symmetry of the model). We recall that, by construction of the training procedure, $E_B^{\text{ref}}/E_T \geq 1$. We observe that the histogram is peaked slightly above the minimal value, which confirms that the tuning procedure produces $E_B^{\text{ref}} \sim E_T$ and $E_R^{\text{ref}} \sim -E_T$. Fig 4C shows the histograms of the number of greedy steps in the tuning procedure, which is

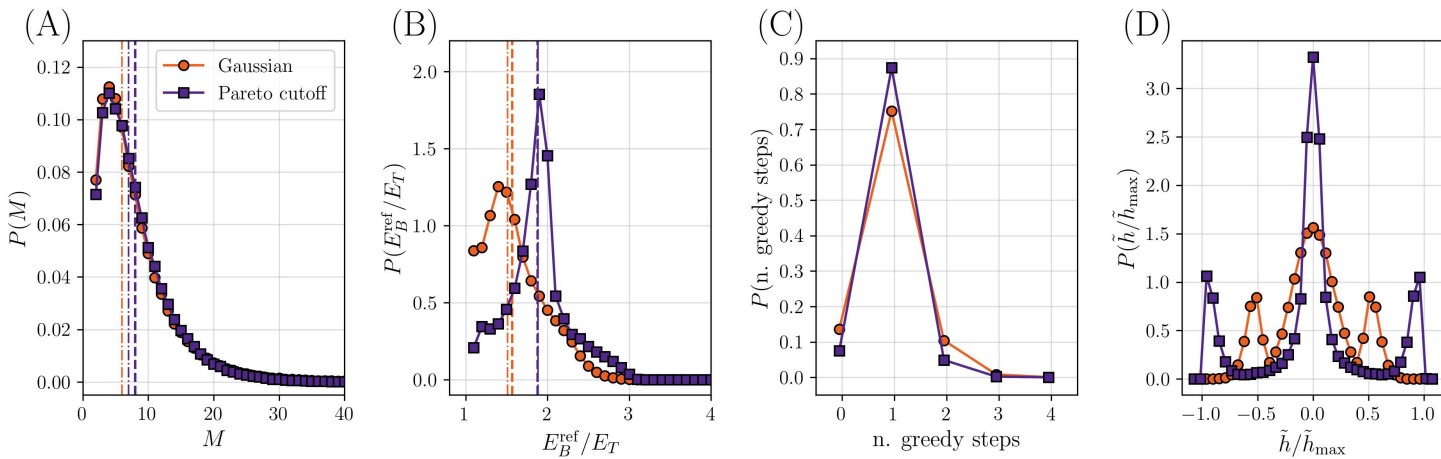

**Fig 4. Distribution of some relevant quantities for two different calibrated models, the Gaussian model and the Pareto cutoff model.** In the former, $L = 500$ SMEs are extracted from a Gaussian distribution with unit variance and the tuning procedure has $p = 0.26$, $E_T = 2.0$; in the latter, $L = 500$ SMEs are extracted from a distribution with Pareto tails decaying with $\alpha = 0.7$ and a cutoff $|h| < 2.0$, and the tuning procedure has $p = 0.25$, $E_T = 1.1$. In each plot the dashed and dot-dashed vertical lines represent the mean and the median of the data, respectively. **(A)** Distribution of the number of mutations. **(B)** Distribution of the value of the positive (blue) reference phenotype divided by $E_T$. **(C)** Distribution of the number of greedy steps needed to reach the target value $E_T$ when generating the two reference variants. **(D)** Distribution of SMEs $\tilde{h}_i$ selected by the tuning procedure to generate the two reference variants.

peaked around one, as expected. Finally, Fig 4D shows the histograms of the $M$ SMEs $\tilde{h}_i$ that are selected by the tuning procedure. These mutations characterize *a posteriori* (after tuning) the space of intermediates between the reference variants, and can be thought of as being those 'fixed' by evolution. We observe a strongly bimodal distribution $P(\tilde{h})$ for both input distributions $P(h)$, which reflects the choice of the tuning procedure: mutations selected by the random steps tend to be small, while mutations selected by the greedy steps tend to be much larger (in a way that can be quantified by extreme value statistics, see the S1 Text).

Next, we analyze the statistical properties of the space of paths connecting the reference variants (Fig 5). We recall that in the calibrated models we find $E_C < E_{\min}^{\text{ref}}$ with probability one. The space of paths thus contains a single genotype

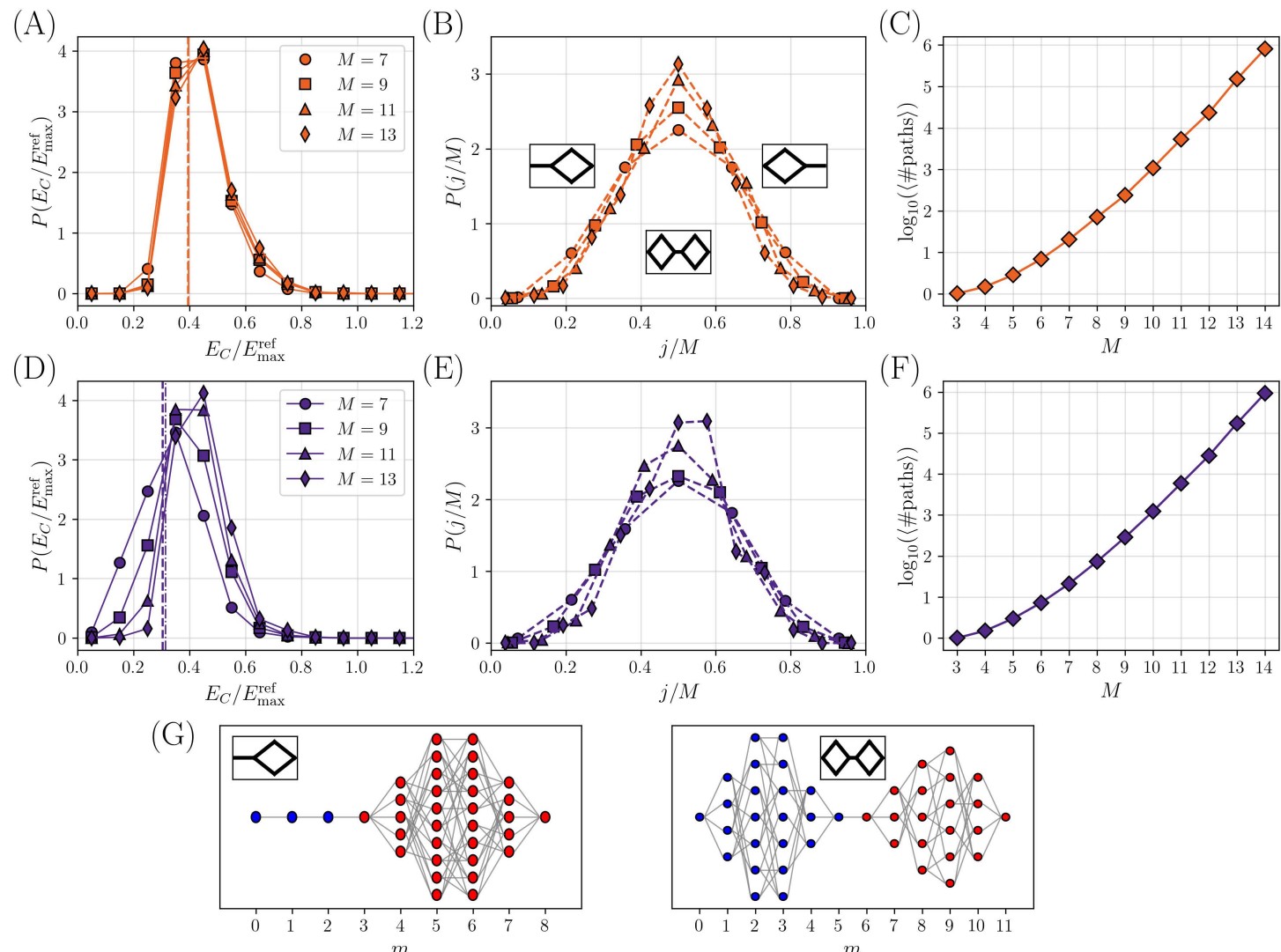

**Fig 5. Statistical properties of the space of paths for two different calibrated model, the Gaussian model (A-C) and the Pareto cutoff model (D-F), with the same parameters as in Fig 4. (A), (D)** Distribution of values of $E_C/E_{\max}^{\text{ref}}$, separately for several values of $M$. The mean and the median (dashed and dot-dashed vertical lines, respectively) are computed with respect to the full distribution (with all $M$ values). **(B), (E)** Probability distribution of the jumper position $j$ along the path, for different values of $M$. **(C), (F)** Average of logarithm of the number of paths that remain viable at $E_C$, as a function of $M$. **(G)** Some examples of topologies, each associated to a schematic representation used in panel **B**.

through which all paths must pass: raising $E_C$ would remove this genotype from the set, and disconnect the reference variants. A bottleneck is thus observed. We note that the jumper genotype, in most cases, is connected by a single mutation to only one genotype carrying the other phenotype, as in Fig 2. However, in a few cases, we also observe situations where the jumper genotype is connected to a few genotypes carrying the other phenotype. In Fig 5A and 5D we show the histograms of $E_C/E_{max}^{ref}$, separately for instances with given $M$. We observe that the histogram is almost independent of $M$ (with more variability observed for the Pareto cutoff model), and peaked around $E_C/E_{max}^{ref} \sim 1/2$ as discussed above. Fig 5B, 5E shows the histogram of the location of the jumper, called $j$, along the mutational path. We observe that the jumper is often located mid-way between the reference variants, leading to a peaked histogram around $j/M \sim 1/2$. The peak (slowly) becomes sharper upon increasing $M$. The qualitative shape of the possible topologies produced by the model is indicated in Fig 5G. Finally, we can compute the number of paths that remain viable at $E_C$. The average of the logarithm of this number is shown in Fig 5C, 5F as a function of $M$. We find that it grows almost linearly at large $M$, indicating that the number of paths grows exponentially in $M$. This suggests that there are exponentially many ways to reach the bottleneck, which could thus maintain a good evolutionary accessibility from both reference sequences [38].

We conclude that our stylized model, which features a global non-linearity on top of an underlying additive phenotype, with properly calibrated parameters, generates topologies that are very close to those observed experimentally in Ref [27] and Sec. 2.1. An example is shown in Fig 2, to be compared with the result obtained re-analyzing the experimental data from Ref [27], shown in Fig 1D. The model generates an ensemble of random fitness functions [5], whose topology is likely (with probability one) characterized by a bottleneck, i.e., by a genotype through which all paths connecting the two reference variants must pass. Of course, the bottleneck can be made less pronounced if one allows for less stringent selection, i.e., reduces $E_C$ [38]. The bottleneck is more likely found mid-way along the path. These features are essentially independent of $M$, i.e., the number of mutations separating the two reference variants, at least in the limited range investigated here.

We note that there is quite a lot of flexibility in the choice of parameters. The initial size $L$ can be varied in a broad interval without affecting much the results (see S1 Text). The tuning parameter $E_T$ has been chosen here in such a way to obtain the desired average of $M$, but other choices are possible. The value of $p$, i.e., the probability of making a greedy step when constructing the reference variants, can be varied in a relatively wide range, provided it is neither too small (otherwise $E_T$ is too small and the reference phenotypes end up overshooting the fitness threshold) nor too large (otherwise $\langle E_C/E_{max}^{ref} \rangle$ decreases). Finally, the *a priori* distribution $P(h)$ that characterizes the input distribution of SMEs can be varied essentially at will in a broad range (see also the S1 Text for other choices of the input distribution). The tuning procedure used to construct the reference variants ensures that the SMEs selected *a posteriori* are broadly distributed, featuring the right balance between almost neutral SMEs and strongly beneficial or deleterious SMEs (Fig 4D), which is a necessary condition for the emergence of a bottleneck. We thus believe that our results are robust and do not depend crucially on the specific choices made during calibration.

## 3 Discussion

In this work, we have demonstrated that functional bottlenecks in protein fitness landscapes can emerge even in the absence of network epistasis, solely due to a non-linear mapping between an underlying additive trait and fitness. To do so, we constructed and analyzed a simple stylized model of global epistasis, in which two reference variants are constructed from a common ancestor by selecting at random either a very beneficial mutation, or a neutral one, with the goal of acquiring two distinct new phenotypes. We have shown that the emergence of bottlenecks separating the two phenotypes is closely tied to the heterogeneity of the single-mutation effects selected during the construction of the reference variants. When mutational effects are heterogeneously enough, evolutionary paths between distinct phenotypes are constrained, leading to the formation of narrow evolutionary corridors.

Our findings provide a novel perspective on how fitness landscape topologies arise and suggest that strong constraints on evolutionary accessibility can exist even in systems governed by simple, global epistatic interactions [2,3,5,19,23,24,27,31,32,36,37]. In fact, in real evolution experiments, one always has access to a single realization of the process. Our stylized model, following and simplifying the approach of Ref [38], constructs an ensemble of fitness landscapes and allows us to assess to what extent the realizations observed in a given experiment are 'typical' of a reasonable ensemble of possibilities. Moreover, despite being stylized, our model successfully captures key qualitative features observed in experimental data, supporting the idea that broad mutational effect distributions play a central role in shaping evolutionary trajectories.

Our results indicate that functional bottlenecks do not necessarily entail network epistasis. Such bottlenecks may instead arise because proteins, under strong evolutionary pressure to acquire new functions, tend to accumulate a few mutations that confer large benefits for the new function, alongside a handful of random neutral mutations introduced by the inherent stochasticity of evolution. Hence, functional bottlenecks may not necessarily indicate complex, higher-order interactions but could instead be a natural consequence of non-linear selection on an underlying additive trait.

Of course, network epistasis does exist in proteins, as it has been shown by a variety of approaches [8,11,12,14,15,20,22,28–30,33–35,44]. Our stylized model might serve as a null model to assess the relevance of network epistasis effects in the analysis of future experimental data.

We also stress that in different kinds of evolutionary dynamics, for instance neutral space evolution under weak selection [61–63], mutations are selected in a different way and our stylized model would likely not apply. The existence of functional bottlenecks during this kind of evolutionary dynamics is less established and might depend on subtle correlations between large groups of interacting mutations [15,64,65]. A recent study of a combinatorial landscape separating naturally occurring bacterial transcription factor binding sites did not find a bottleneck [50].

Our model is simple enough that it could possibly be solved analytically, using techniques from probability and statistical physics [2,5,19]. For instance, the tuning procedure consists in a random walk in the space of mutations, and preliminary results suggest that the probability of the resulting reference traits $E_B^{\text{ref}}$ and $E_R^{\text{ref}}$ can be written in simple form. Making some analytical progress would eliminate the limitation on the number of mutations $M$ separating the two reference variants. Future work could explore this possibility to extend our approach to larger mutational spaces, incorporate additional biological constraints, and investigate the robustness of these results across different fitness functions and evolutionary pressures, possibly with more than two phenotypes.

It should be noted that numerous techniques for extracting genotype-trait-fitness relationships directly from data – including modern machine learning frameworks – have emerged in what is now a rapidly expanding field (see, e.g., [44,66–73]). The present work adopts a fundamentally different approach. Rather than attempting to fit specific empirical datasets, we propose here a simple stylized model, designed to identify relevant qualitative features and universal constraints in a generic way.

By highlighting the role of mutational heterogeneity in the emergence of functional bottlenecks, our study contributes to a deeper understanding of the constraints shaping protein evolution and hopefully opens new directions for theoretical and experimental exploration of evolutionary landscapes.

## 4 Methods

For convenience, we summarize here in closed form the main steps that have to be followed to generate an instance of our model.

### 4.1 Genotype representation and underlying additive phenotype

The first step consists in choosing a value of $L$ and considering a genotype $\mathbf{a} = (a_1, a_2, \ldots, a_L)$ as a binary sequence of length $L$, where each site $i$ can take values $a_i \in \{0, 1\}$. The genotype-dependent underlying additive trait is given by:

$$E(\mathbf{a}) = \sum_{i=1}^{L} h_i a_i,$$

(7)

where the single-mutation effects (SMEs) $h_i$ are independent and identically distributed according to the *a priori* distribution $P(h)$. The two choices we used are a Gaussian distribution with zero mean and unit variance, and a Pareto distribution with cutoff:

$$P(h) = A \begin{cases} 1, & |h| < 0.1, \\ 1/|h|^{\alpha+1}, & 0.1 < |h| < 2, \\ 0, & |h| > 2, \end{cases} \qquad (8)$$

with $\alpha = 0.7$ and the constant $A$ determined by normalization. After sampling, the SMEs are shifted to ensure $\sum_{i=1}^{L} h_i = 0$. These choices fix an additive constant and an overall multiplicative factor in $E(\mathbf{a})$, which can be absorbed by $\beta$ and $E_T$ in the fitness function, without loss of generality:

### 4.2 Fitness function and phenotypic classes

The global epistatic fitness function is a non-linear transformation of $E(\mathbf{a})$, defined separately for the blue (B) and red (R) phenotypes:

$$F_B(E) = \frac{\phi_0}{1 + e^{\beta(E_{th}-E)}}, \quad F_R(E) = \frac{\phi_0}{1 + e^{\beta(E_{th}+E)}}, \qquad (9)$$

where $\phi_0 = 1$ without loss of generality, $E_{th}$ is the functionality threshold parameter, and $\beta \gg 1$ controls the sharpness of the fitness transition. Accordingly, genotypes with $E(\mathbf{a}) > E_{th}$ are functional for the "blue" phenotype, those with $E(\mathbf{a}) < -E_{th}$ are functional for the "red" phenotype, and those with $|E(\mathbf{a})| \ll E_{th}$ are non-functional.

### 4.3 Reference variants construction

The red and blue reference variants, $\mathbf{a}_R$ and $\mathbf{a}_B$, are independently generated starting from the ancestral genotype $\mathbf{a} = (0, 0, \ldots, 0)$ by sequentially introducing mutations. At each step, with probability $p$, the mutation with the largest contribution to $E$ in the desired direction is chosen (greedy step), and with probability $1 - p$, a random site is mutated (random step). The process is iterated until $E_R^{ref} = E(\mathbf{a}_R) < -E_T$ and $E_B^{ref} = E(\mathbf{a}_B) > E_T$.

### 4.4 Mutational paths and bottleneck characterization

The mutational distance $M$ between $\mathbf{a}_R$ and $\mathbf{a}_B$ is defined as the number of differing sites. All $M!$ possible evolutionary paths between the reference variants are considered, and a value $E_C$ is determined such that at least one path of single mutations exists, maintaining $|E(\mathbf{a})| > E_C$ at all intermediate steps. The distribution of $E_C$ and the likelihood of a single critical 'jumper' mutation define the presence and severity of the bottleneck.

## Supporting information

**S1 Text. This supporting document contains all supplementary figures cited in the main text.**
(PDF)

## Acknowledgments

We warmly thank Sid Nagel and Martin Weigt for providing very useful feedback and inspiration all along the development of this work, and Emily Hinds for very useful guidance in reproducing the data analysis from Ref [27].

## Author contributions

**Conceptualization:** Anna Ottavia Schulte, Samar Alqatari, Saverio Rossi, Francesco Zamponi.

**Data curation:** Anna Ottavia Schulte, Saverio Rossi, Francesco Zamponi.

**Formal analysis:** Anna Ottavia Schulte, Saverio Rossi, Francesco Zamponi.

**Funding acquisition:** Francesco Zamponi.

**Investigation:** Saverio Rossi, Francesco Zamponi.

**Methodology:** Anna Ottavia Schulte, Samar Alqatari, Saverio Rossi, Francesco Zamponi.

**Resources:** Francesco Zamponi.

**Software:** Anna Ottavia Schulte, Saverio Rossi, Francesco Zamponi.

**Supervision:** Saverio Rossi, Francesco Zamponi.

**Validation:** Anna Ottavia Schulte, Samar Alqatari, Saverio Rossi, Francesco Zamponi.

**Writing – original draft:** Anna Ottavia Schulte, Samar Alqatari, Saverio Rossi, Francesco Zamponi.

**Writing – review & editing:** Anna Ottavia Schulte, Samar Alqatari, Saverio Rossi, Francesco Zamponi.

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
