## [Decision Letter · Decision Letter 0]

3 Dec 2025

Functional bottlenecks can emerge from non-epistatic underlying traits

PLOS Computational Biology

Dear Dr. Zamponi,

Thank you for submitting your manuscript to PLOS Computational Biology. After careful consideration, we feel that it has merit but does not fully meet PLOS Computational Biology's publication criteria as it currently stands. Therefore, we invite you to submit a revised version of the manuscript that addresses the points raised during the review process.

We look forward to receiving your revised manuscript.

Kind regards,

Marco Cosentino Lagomarsino

Academic Editor

PLOS Computational Biology

Tobias Bollenbach

Section Editor

PLOS Computational Biology

**Additional Editor Comments:**

The manuscript has been reviewed by two senior experts in the field. While their overall enthusiasm on this study differs, they converge on a series of points to address, regarding both the main model assumptions and the comparison with key recent literature. Both reviewers agree that the manuscript addresses an interesting and timely question.

My own view aligns more with reviewer 1, and therefore I will be happy to reconsider the manuscript once all reviewers' remarks have been fully addressed. Note however that additional work on the model and a major revision on the manuscript addressing central claims, relation with literature, and model-data comparisons are required for this revision.

Kind Regards

MCL

**Journal Requirements:**

5) We note that your Data Availability Statement is currently as follows: "This work only used publicly available data. All data are already available in public repositories.". Please confirm at this time whether or not your submission contains all raw data required to replicate the results of your study. Authors must share the “minimal data set” for their submission. PLOS defines the minimal data set to consist of the data required to replicate all study findings reported in the article, as well as related metadata and methods (https://journals.plos.org/plosone/s/data-availability#loc-minimal-data-set-definition).

**Reviewers' comments:**

Reviewer's Responses to Questions

**Comments to the Authors:**

Reviewer #1: The manuscript investigates to what extent a particular fitness landscape structure, where two distinct phenotypic functions are

connected in genotype space through a functional bottleneck consisting of a small number of genotypically accessible paths, can arise purely from global epistasis. The latter term refers to a scenario where a non-epistatic (i.e., linear/additive) genotype-phenotype map is combined with a nonlinear phenotype-fitness map to yield an epistatic (and possibly rugged/complex) genotype-fitness landscape. The broader context of the work is the general question as to whether empirically observed fitness landscape structures can be generated by global epistasis without invoking additional "network" epistasis on the level of the genotype-phenotype map. The authors provide a clear introduction to the context of the work and answer the guiding research question affirmatively by constructing an explicit global epistasis model that produces the desired landscape structure. The paper is clearly written and well organized, and makes an important contribution to our understanding of protein evolution. I recommend publication in PLoS Computational Biology after the following comments have been addressed.

1. My main concern about the manuscript is that it did not become clear to me how surprised I should be by the result. The construction of the toy model involves a lot of deliberate fine-tuning towards the desired bottleneck structure, and it is hard to see how this procedure could have NOT worked. More specifically, at the end of the first paragraph of the Discussion the authors write

"When mutational effects are broadly distributed, evolutionary paths between distinct phenotypes are constrained, leading to

the formation of narrow evolutionary corridors."

which suggests that such corridors will NOT form if the effects are not sufficiently heterogeneous, but (as far as I could see) this

claim is not substantiated by systematically varying the amount of effect size heterogeneity in the model. This central point needs to be clarified.

2. In the discussion of the state of the art in the Introduction, I was surprised that the recent work from the lab of Andreas Wagner

on large-scale high-throughput protein fitness landscapes is not mentioned. In particular, the investigation of the evolution of

bacterial transcription factor binding sites towards three global E. coli transcription factors seems to consider an evolutionary

scenario that is quite similar to the one addressed here (Westmann et al., https://doi.org/10.1101/2024.11.10.620926).

3. At the end of the Introduction the authors claim that "the model is analytically solvable to some extent" and repeat this statement

in the Discussion, but give no hints as to how such an analytic solution could be achieved. They should either provide more details

or omit these statements.

4. Regarding the distinction of the present model from Fisher's geometric model (FGM) mentioned on page 6, I would like to

point out that variants of FGM with multiple phenotypes have been considered previously in the literature (for example,

G. Martin & T. Lenormand, Evolution 69 (2015) 1433–1447).

Reviewer #2: The work entitled “Functional bottlenecks can emerge from non-epistatic underlying traits” by Schulte et al. studies a mutant library data set to better understand epistasis and its effects on functional bottlenecks. The basic question, whether additive underlying trait which maps nonlinearly to fitness (global epistasis) is sufficient to produce functional bottlenecks or whether also interactions explicitly at the level of fitness are needed (network epistasis), is an interesting one. The manuscript tries to address this question by analysing an existing data set of a mutant library (2^13 sequences) with corresponding fluorescent measurements in conjunction of a “toy” model.

Although the research question is of broad interest, the taken approach and the manuscript has substantial issues that make the work a poor match to PLOS CB.

1)The authors mix fitness with trait value “Note that here, fitness refers to fluorescence intensity, which is unrelated to reproductive success.” , possibly this is the norm also by others analysing comparable data but this problem seriously limits the interpretation of the results. In effect, the data allows to study the mapping between genotype and this trait, no more no less. This fundamanetal issues leads to difficulties later e.g. by needing to resort to arbitrariness of thresholds what makes a viabale genotype “As observed in Ref [27] and evident from Fig. 1b, the threshold for defining a genotype as ‘functional’ is somewhat arbitrary, and playing with this threshold changes the topology substantially [38]. ”

2)The “toy” model approach and linking it with data to seem to a large degree arbitrary, “Note that other fitting functions could be used in the analysis, such as the sigmoid function” , “While the results are sensitive to the choice of ϕ, here for simplicity we stick to the choice made in Ref [27], because the analysis of this section serves mostly as a motivation for the rest of the paper.” Or “To capture a few minimal ingredients inspired by the results of Sec. II A and the above discussion, we introduce a toy model featuring only global epistasis, defined as follows: … “ then other to the reader arbitrary choises follow. How is the reader supposed to believe that these choises are reasonable, supported by the data but not over fitting?

3)Overall, the comparison between the model and data is too qualitatively, where are proper model selection, complexity and fit error analysis? Also the authors appear to have taken some parts of the fits from another work “where we used for simplicity the same non-linear function ϕ−1 R (x) = ϕ−1 B (x) = x0.44 that was chosen in Ref [27] to minimize the epistatic contributions.”, which further obscures what has been done here in terms of the fits.

4)There are several very interesting recent developments in the field of trying to understand the structure of genotype to trait to fitness functions which are exploiting modern machine learning methods see e.g. Andreas Wagner, Genotype sampling for deep-learning assisted experimental mapping of a combinatorially complete fitness landscape, Bioinformatics, Volume 40, Issue 5, May 2024, btae317, https://doi.org/10.1093/bioinformatics/btae317

Learning the Shape of Evolutionary Landscapes: Geometric Deep Learning Reveals Hidden Structure in Phenotype-to-Fitness Maps

Manuel Razo-Mejia, Madhav Mani, Dmitri A. Petrov

doi: https://doi.org/10.1101/2025.05.07.652616

Although in general I see value trying to get insight with minimal modelling, I would expect much more comprehensive and less ad-hoc/arbitrary approach to fitting the model to data to find the analysis sound and convincing.

**Have the authors made all data and (if applicable) computational code underlying the findings in their manuscript fully available?**

Reviewer #1: Yes

Reviewer #2: Yes

PLOS authors have the option to publish the peer review history of their article (what does this mean? ). If published, this will include your full peer review and any attached files.

**Do you want your identity to be public for this peer review?** For information about this choice, including consent withdrawal, please see our Privacy Policy .

Reviewer #1: **Yes:** Joachim Krug

Reviewer #2: No

**Figure resubmission:**

**Reproducibility:**



---

## [Decision Letter · Decision Letter 1]

9 Feb 2026

Dear Dr. Zamponi,

We are pleased to inform you that your manuscript 'Functional bottlenecks can emerge from non-epistatic underlying traits' has been provisionally accepted for publication in PLOS Computational Biology.

Best regards,

Marco Cosentino Lagomarsino

Academic Editor

PLOS Computational Biology

Tobias Bollenbach

Section Editor

PLOS Computational Biology

I am glad to report that both reviewers now converged on recommending acceptance, recognizing the value of this study, with Reviewer 2 wishing that the empirical data analysis would have surfaced more to the foreground.

Reviewer's Responses to Questions

**Comments to the Authors:**

Reviewer #1: The revisions have fully clarified the questions raised in my previous report. I recommend publication of the manuscript in its present form.

Reviewer #2: The manuscript has improved with the switch of emphasis towards to modelling/conceptual logic and not aiming to work with the empirical data set in a rigorous way.

**Have the authors made all data and (if applicable) computational code underlying the findings in their manuscript fully available?**

Reviewer #1: Yes

Reviewer #2: Yes

PLOS authors have the option to publish the peer review history of their article (what does this mean? ). If published, this will include your full peer review and any attached files.

**Do you want your identity to be public for this peer review?** For information about this choice, including consent withdrawal, please see our Privacy Policy .

Reviewer #1: **Yes:** Joachim Krug

Reviewer #2: No

---

## [Editor Report · Acceptance letter]

PCOMPBIOL-D-25-01924R1

Functional bottlenecks can emerge from non-epistatic underlying traits

Dear Dr Zamponi,

I am pleased to inform you that your manuscript has been formally accepted for publication in PLOS Computational Biology. Your manuscript is now with our production department and you will be notified of the publication date in due course.

With kind regards,

Judit Kozma
